

# pH-Dependence of Brown Carbon Optical Properties in Cloud Water

Christopher J. Hennigan[1], Michael McKee[1], Vikram Pratap[1], Bryanna Boegner[1], Jasper Reno[1], Lucia Garcia[1], Madison McLaren[1], Sara M. Lance[2]

[1]Department of Chemical, Biochemical and Environmental Engineering, University of Maryland, Baltimore County, Baltimore, 21250, USA
[2]Atmosperic Sciences Research Center (ASRC), University at Albany, Albany, 12226, USA

*Correspondence to*: Christopher J. Hennigan (hennigan@umbc.edu)

**Abstract.** Light-absorbing organic species present in aerosols, collectively called brown carbon (BrC), contribute important but highly uncertain effects on climate. Clouds likely represent a significant medium for secondary BrC production and for bleaching reactions, though the relative importance of formation and loss processes in clouds is unknown at present. The acidity (or pH) of atmospheric particles and clouds affects the optical properties of BrC and bleaching rates. Given the wide variability of pH in the atmosphere (pH in particles and clouds ranges from -1 to 8), the optical properties of BrC and its bleaching behavior are expected to vary significantly, and the link between pH and BrC is yet another uncertainty in attempts to constrain its climate forcing effects. In this work, we characterize the pH-dependence of BrC optical properties – including light absorption at 365 nm ($Abs_{365}$), mass absorption coefficient ($MAC_{365}$), and the absorption Ångström exponent (AAE) – in bulk cloud water sampled from the summit of Whiteface Mountain, NY. In all samples ($n = 17$), $Abs_{365}$ and $MAC_{365}$ increased linearly with increasing pH, highlighting the importance of reporting pH in studies of BrC in aqueous media. There was strong variability in the sensitivity of $Abs_{365}$ to pH, with normalized slopes that ranged from 5.1% to 17.2% per pH unit. The normalized slope decreased strongly with increasing cloud water $[K^+]$, suggesting that the non-biomass burning BrC has optical properties that are more sensitive to pH than BrC associated with biomass burning. AAE also showed a distinct pH-dependence, as it was relatively flat between pH 1.5 – 5, then decreased significantly above pH 5. The cloud water composition was used to inform thermodynamic predictions of aerosol pH upwind/downwind of Whiteface Mountain and the subsequent changes in BrC optical properties. Overall, these results show that,



in addition to secondary BrC production, photobleaching, and the altitudinal distribution, the climate forcing of BrC is quite strongly affected by its pH-dependent absorption.

## 1 Introduction

Light-absorbing organic compounds, or chromophores, in particulate matter are collectively referred to as brown carbon (BrC).  BrC exhibits a strong spectral dependence whereby the absorption efficiency
increases as wavelength decreases (Lack and Cappa, 2010; Kirchstetter et al., 2004).  The light absorbing properties of BrC – both the absorption efficiency at a given wavelength and the wavelength-dependence – vary considerably in the atmosphere (Saleh, 2020).  This variability in optical properties is due to the multitude of different organic compounds that contribute to BrC (Laskin et al., 2015).  Regionally, the radiative forcing of BrC can represent an important component in the direct effect of aerosols on the
radiative balance (Zhang et al., 2017).  However, the global climate forcing attributed to BrC is largely unconstrained at present due to the chemical complexity and widely variable optical properties; global climate models predict a net warming effect from BrC but estimates of the direct radiative forcing vary by almost a factor of 20 (from $+0.03$ W m$^{-2}$ up to $+0.57$ W m$^{-2}$) (Saleh, 2020).

BrC is different from other absorbing aerosols, dust and black carbon (BC), because it has prominent primary and secondary sources (Laskin et al., 2015).  Unlike BC and dust, which are removed from the atmosphere only through wet and dry deposition, it also undergoes chemical losses initiated by oxidants and direct photolysis (collectively termed bleaching), that can rapidly diminish its light absorbing properties (Hems and Abbatt, 2018).  Multiphase atmospheric processes, including those in clouds, play
a key role in the life cycle of BrC, as they can facilitate production or loss, depending on the conditions (Hems and Abbatt, 2018; Laskin et al., 2015; Schnitzler and Abbatt, 2018; Zhao et al., 2015; Lee et al., 2014; Yu et al., 2016; Lin et al., 2015).  Many laboratory studies have investigated BrC formation and loss using cloud water mimics (e.g., Powelson et al., 2014; De Haan et al., 2018) but few using real atmospheric cloud water samples.  This represents a key knowledge gap in understanding the temporal
and spatial distribution of BrC and its effects on climate.





The acidity (or pH) of aerosols and clouds has tremendous importance for numerous atmospheric processes and for associated environmental effects (Pye et al., 2020). This includes an effect of pH on the radiative forcing of BrC compounds. pH affects the light-absorbing properties of many organic compounds in aqueous media (Baes and Bloom, 1990). Aerosol samples collected in the southeastern U.S. showed pronounced pH-dependent absorbance spectra (Phillips et al., 2017). This was observed for both background ambient samples and those that had been influenced by biomass burning emissions. The increase in absorbance with increasing pH suggests that carboxylic acids and phenols contributed significantly to BrC in this study (Phillips et al., 2017). A similar study found that the aqueous extracts of biomass burning aerosols exhibited a strong spectral dependence with pH (Cai et al., 2018). Further, pH had an important effect on the evolution of BrC bleaching in photolysis experiments conducted with the aqueous aerosol extracts (Cai et al., 2018). This is expected because many aqueous reaction rates are pH-dependent (Pye et al., 2020; Tilgner et al., 2021).

The pH of atmospheric particles, cloud, and fog droplets spans a wide dynamic range, from approximately -1 in highly acidic particles up to 8 in clouds containing alkaline components (Shah et al., 2020; Pye et al., 2020). Therefore, the optical properties of BrC and its bleaching behavior are expected to vary significantly in the atmosphere, as well. The pH-dependence of BrC absorbance and bleaching may be especially dynamic in cloud cycles because the pH of CCN particles can increase by 3-4 pH units when they activate and form cloud droplets or decrease by the same amount when they transition from cloud droplets back to aqueous particles (Rusumdar et al., 2020). This suggests that the light absorbing properties of chromophores may be dramatically different in clouds compared to the properties of the same compounds in aqueous haze particles. In addition to in-cloud BrC transformations due to secondary formation and bleaching, changes in BrC absorbance with pH need to be characterized in order to understand the impact of clouds on BrC.

The purpose of this study is to characterize the pH-dependence of BrC optical properties in cloud water samples collected at Whiteface Mountain, NY (WFM). The samples were collected over two summers and have diverse chemical composition and back trajectories, suggesting a variety of source influences



85 and airmass ageing. The analysis of BrC optical properties is paired with aerosol thermodynamic equilibrium modeling and cloud water composition measurements to calculate the aerosol liquid water content and pH of aerosols upwind of WFM. Using the measured pH-dependence of BrC optical properties, the modeled aerosol pH upwind of WFM informs changes to BrC radiative forcing in the atmosphere.

## 2 Materials and Methods

Cloud water samples were collected at the summit of Whiteface Mountain, NY (latitude N44°21′58″ and longitude W 73°54′10″, 1483 m a.s.l.) during the summers of 2018 and 2019. The site has been used continuously for atmospheric chemistry and cloud research for more than three decades (Mohnen and Kadlecek, 1989). Procedures for cloud water collection and chemical analysis have been described in detail before (Lance et al., 2020; Schwab et al., 2016; Lawrence et al., 2023). Briefly, cloud water collection occurred via a passive Mohnen omni-directional sampler when non-precipitating liquid clouds were present under select meteorological conditions. The collected cloud water was refrigerated (4°C), and analyzed for inorganic ionic species, organic acids, pH, conductivity, and total organic carbon (TOC) content. A subset of the samples was syringe filtered (0.45 μm) prior to analysis and thus, the analyzed organics represents WSOC instead of TOC (Lance et al., 2020). The remaining sample volume was frozen (-20°C) for further analysis. Some of these frozen samples were packed in dry ice and shipped overnight to the University of Maryland, Baltimore County for BrC analysis.

In this study, BrC is operationally defined as the water-soluble organic carbon compounds that absorb light in the 300 – 500 nm wavelength range. Many chromophores are found in the atmosphere that are insoluble or are soluble in other solvents (e.g., methanol) (Jiang et al., 2022; Zeng et al., 2020; Zhang et al., 2013). For the cloud water samples that were filtered, and since no additional organics were added, our analyses neglect any chromophores that are insoluble in water or soluble in other solvents (e.g., methanol), which can be 40-50% of BrC globally (Zeng et al., 2020). Absorbance spectra of cloud water samples at each pH were recorded based on the method described previously (Pratap et al., 2020). A liquid waveguide capillary cell (LWCC, World Precision Instruments) with 50 cm path length was





coupled via fiber optic cables to a light source (200 – 1600 nm, Ocean Insight DH-mini) and two spectrometers, one to record absorbance spectra (FLAME-S, Ocean Optics) and one to continuously monitor the light source stability (STS-VIS, Ocean Insight). An automated syringe pump (model C3000, TriContinent Scientific) delivered the pH-adjusted cloud water sample to the LWCC, which was thoroughly rinsed with DI water (>18.2 MΩ·cm) in between each sample. Absorbance spectra from 300 – 500 nm were recorded every 3 seconds and averaged over 4 minutes. Mass absorption coefficients at 365 nm ($MAC_{365}$, $m^2 \, g^{-1}$) were calculated according to Saleh (2020). The absorbance spectra from 300 – 500 nm were fit with an exponential function (Igor Pro, WaveMetrics) to calculate the absorption Ångström exponent at each pH ($AAE_{pH}$).

The optical properties of each cloud water sample were measured at pH from 11 to 1.5. The cloud water sample was first adjusted to pH 11 using 1.0 M NaOH (Fisher Scientific). Downward pH adjustments of ~1 pH unit were made through additions of 0.1 M and 3.0 M HCl (Fisher Scientific). pH was measured using an Orion Star A211 meter with Ross Ultra pH Electrode (Orion 8103BNUWP). The actual pH after NaOH and HCl additions was variable and depended on the cloud water composition (**Table 1**). Therefore, the actual pH at each step was recorded and used for the analyses. After the pH stabilized, approximately 1 mL of the cloud water sample was injected into the LWCC to fill the LWCC sample loop and tubing void volumes. The optical measurements accounted for the small diluting effects of the NaOH and HCl additions. Initially, these pH titrations and optical analyses were performed in duplicate on each cloud water sample; however, a high degree of repeatability was observed (**Fig. 1**), so subsequent samples were only analyzed once to conserve the limited sample volumes. QA/QC on the experimental procedure was run using the full range of pH adjustments to DI water. The DI water at all pH levels showed absorbance values approximately 2 orders of magnitude lower than the cloud water samples and often below the system limit of detection: No background adjustments were made to the cloud sample measurements. Cloud water samples were also compared to Suwannee River Natural Organic Matter (SR-NOM, International Humic Substances Society), a well-characterized material that has similarities to BrC sampled in the atmosphere (Green et al., 2015; Graber and Rudich, 2006).



Back trajectories for each cloud water sample were calculated using the Hybrid Single-Particle Lagrangian Integrated Trajectory model (HYSPLIT, (Stein et al., 2015)). Following the approach of Lawrence et al. (2021), 6-day back trajectories were initiated at each hour of cloud water sampling using meteorology from the North American Regional Reanalysis (NARR) 32 km x 32 km. Output from the HYSPLIT model included the air mass location and altitude, as well as meteorological parameters. The

temperature (T) and relative humidity (RH) output from HYPLIT were used as inputs to the ISORROPIA-II aerosol thermodynamic equilibrium model (Fountoukis and Nenes, 2007). The inorganic ionic composition of the cloud water samples was used for the ISORROPIA-II model input, with the exception that $Ca^{2+}$ and $Mg^{2+}$ concentrations were excluded because a decadal analysis of WFM cloud composition revealed that these species likely derive predominantly from coarse particles (Lawrence et al., 2023). To

provide an estimate of fine aerosol pH along the back trajectory, ISORROPIA-II was run in forward mode with solids formation suppressed (i.e., metastable) according to Pye et al. (2020). We note that this approach assumes a constant chemical composition along the entire 6-day back trajectory, which is clearly not realistic. However, this exercise shows pH variations along the back trajectory that may arise solely from the changes in T and RH as airmass altitude and position changes. In many locations, T and RH

exert a greater influence on fine aerosol pH than composition, like sulfate and ammonium (Battaglia et al., 2017; Zheng et al., 2020; Tao and Murphy, 2021). Therefore, this exercise, while not accurate in predicting the actual fine aerosol pH along the back trajectory, does reflect the magnitude of pH variability that may be expected as airmasses were transported to WFM and how they compared to the cloud water pH.

## 3 Results

An overview of the cloud water samples analysed for this study is presented in **Table 1**. The cloud liquid water content ranged from 0.26 g m$^{-3}$ to 0.87 g m$^{-3}$ (median 0.53 g m$^{-3}$) and the temperature ranged from 7.2 °C to 21.0 °C (median 17.2 °C). Cloud water composition also showed quite a bit of variability: pH ranged from 4.27 to 6.41 (median 4.70). TOC concentrations in the cloud water varied by more than an

order of magnitude, from 106 μM to 1770 μM (median 505 μM). Amongst the inorganic ionic species measured, $NH_4^+$ had the highest median concentration (58.3 μM), followed by $NO_3^-$ (32.5 μM), $SO_4^{2-}$



(22.0 µM), and Cl⁻ (19.0 µM). In a prior study at WFM, Cook et al. (2017) found that $K^+$ concentrations exceeded 0.04 mg L⁻¹ (1 µM) in cloud water samples influenced by fire emissions. In the present study, 12 of the 17 samples had $K^+$ concentrations above 0.04 mg L⁻¹, suggesting fire emissions had an important
impact on cloud water composition. This is supported by the linear correlation between $K^+$ and TOC concentrations ($R^2$ = 0.62) in our samples and prior studies at WFM observing frequent influence of fire emissions on cloud water samples during the summer (Lance et al., 2020; Lee et al., 2022).

The distribution of $MAC_{365}$ values at pH 4 is shown in **Figure 2**. $MAC_{365}$ values at pH 4 ranged from
0.12 m² g⁻¹ up to 0.68 m² g⁻¹ (median of 0.34 m² g⁻¹). The median $MAC_{365}$ value for WFM cloud water samples was similar to the value observed for Suwannee River NOM (0.30, green hatched bar in Fig. 2). A key feature of the BrC optical properties in our samples was the strong pH-dependence of absorption, as illustrated in Fig. 1 for duplicate analyses of sample 1922801. All samples showed a positive linear relationship, as absorption (and MAC) increased with increasing pH. This result is consistent with
observations of BrC in ambient aerosols sampled in Georgia, including under conditions of biomass burning influence (Phillips et al., 2017).

Although a positive linear relationship between $MAC_{365}$ and pH was observed in all samples, there was considerable variation in the slope of this relationship for different cloud water samples (**Fig. 3** and **Table**
**2**). Each sample's absorbance at 365 nm ($Abs_{365}$) at each pH was normalised to the sample $Abs_{365}$ at pH 1.5. The slope of the normalized $Abs_{365}$ vs. pH is a measure of how much BrC absorption changes for each pH unit change. The lowest relative slope was observed for sample 1822702 (slope of 0.051), which indicates the $Abs_{365}$ varied by 5.1% per pH unit. Thus, a change of 6 pH units would only change the BrC absorption at 365 nm for this sample by ~30%. The highest relative slope was observed for sample
1920702 (17.2% change in $Abs_{365}$ per pH unit), where a 6 pH unit change would result in greater than a factor of two change for the BrC absorption at 365 nm. The results in **Fig. 3** are contrasted with results from Phillips et al. (2017), where slopes of 8% and 13% per pH unit were observed for BrC sampled in ambient aerosols uninfluenced and significantly influenced by biomass burning, respectively (solid red and green lines in **Fig. 3**). The sensitivity of $Abs_{365}$ to pH observed in WFM samples was similar to that





of SR-NOM (thick dashed black line in **Fig. 3**), which had a slope of 10.7% per pH unit. The greater variability observed in our study may be due to the ageing of air masses sampled at WFM and more diverse source influences, which is discussed in detail below.

Variability in the response of BrC absorption to changes in pH was likely influenced by BrC sources.
**Figure 4** shows the relationship between the normalized slope from Fig. 3 and the cloud water $K^+$ concentration. There was a clear decreasing trend in the sensitivity of BrC absorption to pH as $[K^+]$ increased. As discussed above, $[K^+]$ in WFM cloud water samples was used to identify biomass burning influence (Cook et al., 2017). The results in **Figure 4** suggest that aged biomass burning samples have less sensitivity to pH than ambient samples uninfluenced (or only lightly influenced) by biomass burning.
This contrasts the results of a prior study that observed samples heavily influenced by fresh biomass burning emissions showed higher sensitivity of BrC absorption to pH (Phillips et al., 2017). The apparent discrepancy between the results in Fig. 4 and those in Phillips et al. (2017) is likely due to atmospheric ageing. Phillip et al. (2017) sampled under conditions when large fires were burning relatively close to the sampling site, suggesting that the emissions were fresh. By contrast, fires burning in the western U.S.
and Canada undergo transport of days (typically 3 – 7 days) before sampling at WFM (Cook et al., 2017; Lance et al., 2020). Changes in BrC optical properties as biomass burning emissions age in the atmosphere are well documented (Saleh et al., 2013; Laskin et al., 2015; Forrister et al., 2015). Photobleaching and secondary BrC production alter the MAC and AAE of primary biomass burning emissions, though the timescale for these changes varies widely in the atmosphere and remains a major
uncertainty in better representing BrC in global models (Saleh, 2020). The present results suggest that one such change not previously reported is that atmospheric ageing reduces the sensitivity of biomass burning BrC optical properties to pH.

Similar to the $MAC_{365}$, the absorption Ångström exponent showed wide variability across the cloud water
samples (**Fig. 5** and **Table 2**). The $AAE_5$ ranged from 4.63 (sample 1822702) to 7.09 (sample 1818204). $AAE_5$ values of cloud water samples were systematically higher than the SR-NOM, which had an $AAE_5$ of 3.39 (green hatched bar in **Fig. 5**). $AAE_5$ values indicate three samples (1822702, 1920004, and





1817601) were 'moderately absorptive' (2.5 < AAE < 5), while the remainder were 'weakly absorptive' (5 < AAE < 8) according to the optical bins defined by Saleh (2020). The AAE values also showed a distinct pH dependence. **Figure 6** shows the mean AAE values for WFM cloud water samples as a function of pH. For pH 1.5 – 5, there was not much variation in AAE with pH; however, AAE decreased significantly above pH 7. The results in Figure 6 are consistent with the results of Qin et al. (2022b), who characterized water-soluble BrC in Beijing and observed a non-linear relationship between AAE and pH, with significant reductions in AAE at pH > 6. This is likely due to chromophores with pKa values above 6. For weakly acidic and moderately basic BrC chromophores, changes in pH below pH 6 impart only minor changes to the acid-base equilibrium and thus the spectral dependence on pH. Above pH 6, BrC speciation becomes more sensitive to pH variations, resulting in changes to AAE in this range.

The optical properties of BrC are highly variable, reflecting the chemical diversity of atmospheric organic compounds that absorb light (Laskin et al., 2015). In general, as BrC becomes more highly oxidized (and thus water-soluble), BrC absorption efficiency weakens and AAE increases (Saleh, 2020). The optical properties of BrC in WFM samples were broadly consistent with the findings in other locations. For example, the AAE values of WSOC sampled in cloud water at Mt. Tianjing, China ranged from 5.37 – 6.31 during a study where cloud water composition was strongly influenced by biomass burning (blue hatched bar in **Fig. 5**, (Guo et al., 2022)). Water-soluble BrC extracted from PM$_{2.5}$ samples exhibited average AAE values of 6.6 – 6.8 in Beijing and 4.8 – 7.3 in other cities in China (Qin et al., 2022a; Wu et al., 2020), while water-soluble PM$_{2.5}$ extracts at various locations in India had AAE values of 5.1 – 5.3 (Kirillova et al., 2014; Srinivas et al., 2016). In the U.S., water-soluble organic carbon had mean AAE values of 7.6 in Los Angeles (Zhang et al., 2013) and 6.0 – 8.3 in Atlanta, depending on the time of year (Hecobian et al., 2010). In the Atmospheric Tomography Missions (ATom-2, ATom-3, and ATom-4) aircraft studies probing global aerosol compositions, water-soluble BrC exhibited mean AAE values of 4.1 – 6.5 (red patterned bars in **Fig. 5**; (Zeng et al., 2020)). There was not an apparent trend in the geographic distribution of AAE, though there was an altitudinal dependence, with lowest AAE values observed at the highest (10 – 13 km) and lowest altitudes (< 1 km) (Zeng et al., 2020).



A comparison of the present results to other studies brings up a critical point: Most of the prior studies did not report the pH of the aqueous extracts or the pH-dependence of absorption or AAE. The results in Figures 1, 3, and 6 show that the optical properties of atmospheric water-soluble BrC depend strongly on pH. Therefore, measurements of BrC in aqueous environments need to include and report pH in order to facilitate interstudy comparisons and to assess the climate forcing effects of BrC.

In addition to the changes in $Abs_{365}$ (and $MAC_{365}$) and AAE with pH, the relative absorption changed with wavelength at each pH, as well. **Figure 7** shows the absorption at a given pH relative to the absorption at pH 1.5 averaged over all cloud water samples. There was a clear wavelength-dependence of the absorption enhancements with increasing pH. At pH < 6, there was a minor increase in absorption from 300 nm up to 340 nm. The mean absorbance spectra below pH 6 were relatively flat from 340 nm up to 400 nm, declined gradually to a minimum around 450 nm, then gradually increased up to 500 nm. Above pH 7, mean absorbance spectra showed a steady increase from 300 nm – 380 nm, peak absorption relative to pH 1.5 at 380 – 400 nm, followed by a similar decline and minimum at ~450 nm. Comparison to other studies reveals both similarities and differences in the wavelength-dependent variations with pH shown in **Fig. 7**. For example, in a study characterizing the effects of pH on water-soluble BrC aerosols in Beijing, there was a similar wavelength-dependence to the absorption enhancement we observed in **Figure 7** (Qin et al., 2022a); however, the magnitude of the absorption enhancement at 400 nm relative to pH 1.5 was higher in WFM cloud water at each pH level.

## 4 Discussion

Our results have important implications for understanding BrC in the atmosphere. It is imperative that studies reporting the optical properties of water-soluble BrC – whether in aerosol extracts, cloud, or fog water – also report the pH at which the measurements were conducted. The *in situ* aerosol pH (ranging from -1 to 8) may be dramatically different from the pH of dilute aqueous extract solutions (most likely in the range of pH 4 to 6). This will facilitate more accurate comparisons between studies. Cloud cycling, with the accompanying 2 – 3 pH unit step changes, results in BrC absorption efficiency changes of 10%





– 50% for the WFM samples. Particle composition and changes in meteorology also change aerosol pH, such that BrC compounds may experience variability of 5 - 6 pH units over their atmospheric lifetime.

Our results suggest that these pH changes dramatically change the climate forcing effects of BrC. Aerosols are usually more acidic than clouds, so BrC exhibits stronger absorption in clouds than in aerosols. Therefore, in addition to factors that have been considered, such as the altitudinal distribution of BrC (Zhang et al., 2017), accurately constraining the climate forcing effects of BrC requires accounting for the sensitivity to pH.


The variability in BrC absorption due to pH is illustrated in **Figure 8**, which shows back trajectories corresponding to each cloud water sample analysed in this study. The 6-day back trajectories are coloured by the $Abs_{365}$ along the trajectory normalized to the $Abs_{365}$ for cloud water conditions. The relative $Abs_{365}$ changes in Figure 8 are based upon the changes in pH along the back trajectories as well as the sensitivity

of absorption to pH for each sample (normalized slope in Table 2). For all samples, the maximum $Abs_{365}$ occurs in-cloud because the pH is systematically higher than in particles due to the effect of dilution. The predicted changes range from minor to major, depending on the sample. The mean normalized $Abs_{365}$ among all trajectories was 0.637 (median 0.661), indicating a ~1/3 reduction in BrC absorption in the aerosols relative to the cloud water. As described above, aerosol pH modelled along the back trajectory

was computed using the cloud water composition so it only accounts for changes in T and RH. Although these factors have a major effect on pH (Tao and Murphy, 2019), the actual pH along the back trajectories will differ due to changes in aerosol composition. $Abs_{365}$ will further change if secondary BrC production or bleaching occurs during transport to WFM. Therefore, the results in **Fig. 8** serve only as a guide to illustrate the magnitude of BrC absorbance changes that may occur in the atmosphere due to changes in

pH associated with changes in T and RH. **Figure 8** also illustrates the diversity of sources influencing the cloud water samples, consistent with the variable cloud water composition discussed above. The trajectories collectively show marine, clean arctic, biogenic, and polluted continental air mass origins. We attempted to correlate BrC optical properties with the trajectory analyses; however, this was inconclusive because of the small number of trajectories originating in some regions.




These results also inform measurements of BrC that are not conducted in aqueous matrices. For example, experimental approaches such as cavity ringdown spectroscopy, photoacoustic spectrometry, and aethalometry are frequently used to measure total BrC, not just the water-soluble fraction (Washenfelder et al., 2013; Liu et al., 2015). BrC compounds are likely to encounter liquid water during much of their
time in the atmosphere and water-soluble chromophores can dissolve or partially dissolve in the aqueous fraction (Zhang et al., 2012; Pye et al., 2018). Therefore, the optical properties of BrC are not static, but evolve in space and time. Secondary BrC formation and photobleaching contribute to these dynamic changes. Our results show that changing pH, which can be due to changing inorganic aerosol composition, changing T and RH, also contribute to changing BrC optical properties that may be as - or
more - impactful as these other processes.

Our results indicate that pH is a key parameter that may dramatically change the radiative forcing of water-soluble BrC relative to insoluble BrC. Water-soluble BrC makes a variable contribution to the overall light absorption. Guo et al. (2022) found that water-soluble BrC contributed approximately half
of the $Abs_{370}$ in cloud droplet residuals sampled at Mt. Tianjing, China, and Zeng et al. (2020) estimated that the water-soluble fraction of BrC was approximately half of the total BrC in the ATom aircraft studies characterizing the global distribution of aerosol composition and concentration. However, our results show that changes in aerosol and cloud water pH will clearly change the climate forcing of water-soluble BrC relative to the total. Changes in composition, aerosol and cloud processes (e.g., cloud activation or
drying), and meteorology (temperature and RH) strongly affect pH (Pye et al., 2020). Therefore, the contribution of water-soluble BrC to total BrC radiative forcing will change in time and space, as well.

The WFM cloud water samples showed similarities with Suwannee River NOM. Specifically, the MAC at pH 4 (Fig. 2) and the normalized slope of $Abs_{365}$ vs. pH (Fig. 3) for SR-NOM were both very close to
the median WFM cloud water samples. More broadly, Schendorf et al. (2019) characterized the pH effects on optical properties of five different humic substances isolated from aquatic and terrestrial systems and found similar behaviour to our cloud water samples, as absorption increased with increasing pH, though the wavelength-dependence and spectral characteristics differed between different samples.



Our WFM cloud water samples were most similar to their analysis of Suwannee River fulvic and humic
acids (Schendorf et al., 2019), materials that contain hydrophobic organic acids isolated from the SR-
NOM we analysed (https://humic-substances.org/, last accessed 24-Feb-2023).  It is now well known that
chemical and physical characteristics of atmospheric organic matter – especially the light-absorbing
fraction – bear similarities to organic material found in aquatic and terrestrial systems (Kalberer et al.,
2004; Graber and Rudich, 2006).  The present results build upon this body of work and support the use
of these materials, which are abundant and commercially available, as surrogates for further studies into
the pH-dependent optical properties of atmospheric BrC.

Turnock et al. (2019) analysed changes in aerosol radiative forcing due to cloud water pH.  Their study
considered the effects of pH on the aqueous oxidation of $SO_2$ in clouds and how this changed global
particle size distributions.  Ultimately, their model predicted stronger aerosol radiative forcing at higher
cloud water pH because of enhanced $SO_2$ oxidation to sulphate in clouds and the resulting reduction in
gas-phase production of $H_2SO_4$ (Turnock et al., 2019).  Our results show that changes in cloud water pH
also affect the climate forcing of aerosols through effects on BrC absorption.  Increasing pH increased
the absorption efficiency of BrC for all cloud water samples analysed.  Because BrC absorption is
sensitive to pH, photolysis rates (i.e., photobleaching) are sensitive to pH, as well.  For example, Zhao et
al. (2015) observed a factor of ~2 increase in the photolysis rate of a nitrophenol (4-nitrocatechol) going
from pH 3 to pH 5.  Another study observed an increase in the photolysis rates of nitrophenols (guaiacol,
catechol, and 5-nitroguaiacol) with decreasing pH (Yang et al., 2023).  Therefore, cloud and aerosol pH
affect the lifetime of BrC compounds, which further affects their climate forcing.

Finally, our measurements were conducted on liquid cloud samples, so these results are limited to aqueous
environments.  Aerosol liquid water, while often abundant, is also highly variable in the atmosphere
(Nguyen et al., 2016).  Aqueous aerosols contain orders of magnitude less water than cloud droplets, so
organic solutes rapidly transition between highly concentrated and dilute environments during cloud
cycling.  Organic compounds in atmospheric particles span a very wide range of water solubilities,
including compounds measured in the water-soluble fraction (i.e., as WSOC) (Psichoudaki and Pandis,



2013). Changes in liquid water content cause organic aerosols to undergo various phase transitions, including liquid-liquid phase separation and transition to highly viscous "glassy" states (Reid et al., 2018). Acidity can be defined in non-aqueous media, though the pH scale as a measure of solution acidity does
not directly transfer between different solvents (Himmel et al., 2018). Therefore, our results do not inform the optical properties of BrC in non-aqueous environments; however, future studies should address this question. These results reflect the optical properties of organic compounds dissolved in cloud water at their time of sampling under the environmental conditions given in Table 1; a different distribution of solubility would likely change the measured optical properties, as well.


## Data Availability

All data presented in this work are published in Tables 1 and 2. Any data not published, e.g., raw spectra, are available upon request.

## Author Contributions

C. Hennigan: Conceptualization, Methodology, Resources, Funding acquisition, Supervision, Writing – Original Draft, Writing – Review & Editing. M. McKee: Methodology, Validation, Formal analysis, Writing – Review & Editing, Visualization. V. Pratap: Methodology, Validation, Formal analysis, Writing – Review & Editing, Visualization. B. Boegner: Methodology, Validation, Formal analysis,
Writing – Review & Editing, Visualization. J. Reno: Methodology, Validation, Formal analysis, Writing – Review & Editing, Visualization. L. Garcia: Methodology, Validation, Formal analysis, Writing – Review & Editing, Visualization . M. McLaren: Methodology, Validation, Formal analysis, Writing – Review & Editing, Visualization. S. Lance: Methodology, Resources, Writing – Original Draft, Writing – Review & Editing.


## Competing Interests

The authors declare no competing financial interests.

## Acknowledgements



We acknowledge Paul Casson, Richard Brandt, and Eric Hebert for helping set up and service the cloud water collection system.  We acknowledge Dan Kelting and Liz Yerger, Paul Smith's College, for conducting the chemical analysis of the cloud water samples.

**Financial Support**

This work was supported by the U.S. Department of Energy, Office of Biological & Environmental Research (BER) through grant DE-SC0022049.  Cloud water and meteorological measurements reported in this paper were supported by the New York State Energy Research and Development Authority (NYSERDA) Contract 124461. NYSERDA has not reviewed the information contained herein, and the opinions expressed in this report do not necessarily reflect those of NYSERDA or the State of New York.






**Table 1:** Overview of the cloud water composition for samples collected from the summit of Whiteface Mountain and analysed for this study.

| Sample ID | Date | Cloud sampling duration (hr) | Cloud LWC (g/m³) | Temp. (°C) | pH | TOC (μmol/L) |
|---|---|---|---|---|---|---|
| 1817601 | 25-Jun-2018 | 4.7 | 0.51 | 7.2 | | 439.3 |
| 1818204* | 1-Jul-2018 | 10.9 | 0.66 | 19.0 | 4.51 | 988.3 |
| 1818205* | 1-Jul-2018 | 3.8 | 0.33 | 19.0 | 4.57 | 1770 |
| 1820401 | 23-Jul-2018 | 4.95 | 0.40 | 17.7 | 4.60 | 206.1 |
| 1820702 | 26-Jul-2018 | 6.3 | 0.75 | 15.2 | 4.64 | 106 |
| 1821005 | 29-Jul-2018 | 11.1 | 0.87 | 11.5 | 4.88 | 216 |
| 1821301 | 1-Aug-2018 | 7.4 | 0.36 | 15.2 | 4.27 | 498.2 |
| 1822701 | 15-Aug-2018 | 2.1 | 0.55 | 17.2 | 4.62 | 869.2 |
| 1822702 | 15-Aug-2018 | 2.1 | 0.55 | 17.2 | | 732.1 |
| 1822901 | 17-Aug-2018 | 5.8 | 0.72 | 15.8 | 4.76 | 311.4 |
| 1823805 | 26-Aug-2018 | 7.4 | 0.78 | 13.8 | 5.35 | 801.8 |
| 1920004* | 19-Jul-2019 | 11.3 | 0.57 | 18.9 | 5.18 | 512.3 |
| 1921102* | 30-Jul-2019 | 1.9 | 0.27 | 21.0 | 6.41 | 1323.3 |
| 1922202* | 10-Aug-2019 | 10.8 | 0.50 | 15.0 | 5.38 | 344.6 |
| 1922801* | 16-Aug-2019 | 6.8 | 0.51 | 18.3 | 4.28 | 540 |
| 1926801 | 25-Sept-2019 | 2.9 | 0.26 | 16.7 | 5.18 | 385 |

| Sample ID | [Na⁺] (μM) | [K⁺] (μM) | [NH₄⁺] (μM) | [Ca²⁺] (μM) | [Mg²⁺] (μM) | [Cl⁻] (μM) | [NO₃⁻] (μM) | [SO₄²⁻] (μM) |
|---|---|---|---|---|---|---|---|---|
| 1817601 | 0.9 | 1.0 | 20.0 | 1.0 | 0.4 | 25.7 | 11.3 | 14.9 |
| 1818204* | 5.7 | 2.0 | 510.0 | 23.0 | 5.3 | 46.8 | 93.5 | 67.0 |
| 1818205* | 10.0 | 3.1 | 908.9 | 28.2 | 8.2 | 57.3 | 125.0 | 102.4 |
| 1820401 | 31.3 | 1.3 | 14.4 | 3.5 | 2.5 | 28.5 | 21.3 | 10.5 |
| 1820702 | | 0.8 | 20.0 | 2.0 | 0.8 | 13.8 | 24.4 | 9.9 |
| 1821005 | 1.3 | 0.5 | 10.6 | 3.2 | 1.2 | 22.6 | 6.1 | 6.7 |
| 1821301 | 13.9 | 1.5 | 52.2 | 7.5 | 2.1 | 22.3 | | 21.8 |
| 1822701 | 7.0 | 3.1 | 141.1 | 21.5 | 4.9 | 18.9 | 40.5 | 22.3 |
| 1822702 | 0.4 | 2.6 | 80.0 | 19.2 | 4.9 | 19.2 | 35.6 | 21.4 |



| 1822901 | 1.3 | 2.3 | 60.0 | 13.0 | 3.7 | 0.8 | 37.7 | 25.2 |
| 1823805 | 13.0 | 2.6 | 141.7 | 39.2 | 13.6 | 3.7 | 69.7 | 32.8 |
| 1920004* | 2.6 | 1.8 | 38.9 | 11.5 | 2.5 | 24.3 | 25.8 | 13.1 |
| 1921102* | 3.0 | 4.6 | 248.3 | 94.6 | 11.9 | 1.7 | 158.5 | 85.7 |
| 1922202* | 2.6 | 1.8 | 56.7 | 10.2 | 2.5 | 9.9 | 29.4 | 22.9 |
| 1922801* | 2.6 | 2.0 | 96.7 | 10.7 | 4.1 | 9.9 | 90.6 | 30.3 |
| 1926801 |  |  | 9.4 |  |  | 5.4 | 10.6 | 7.9 |

*Denotes samples that were syringe-filtered (0.45 μm); thus the TOC reported is actually WSOC for these samples.

**Table 2:** Summary of key optical properties for cloud water samples.

| Sample ID | $MAC_{365}$ $(m^2\ g^{-1})$ | *Normalized slope $Abs_{365}$ vs. pH | $AAE_2$ | $AAE_5$ | $AAE_9$ |
|---|---|---|---|---|---|
| 1817601 | 0.220 | 0.171 | 5.49 | 4.96 | 4.53 |
| 1818204 | 0.121 | 0.066 | 7.59 | 7.09 | 4.43 |
| 1818205 | 0.123 | 0.070 | 6.86 | 6.50 | 4.81 |
| 1820401 | 0.473 | 0.146 | 5.66 | 6.16 | 6.22 |
| 1820702 | 0.678 | 0.172 | - | - | - |
| 1821005 | 0.486 | 0.109 | 5.94 | 6.41 | 5.01 |
| 1821301 | 0.122 | 0.154 | 6.20 | 5.79 | 6.08 |
| 1822701 | 0.575 | 0.058 | 6.23 | 6.00 | 5.19 |
| 1822702 | 0.427 | 0.051 | 4.53 | 4.63 | 4.77 |
| 1822901 | 0.678 | 0.093 | 6.13 | 5.76 | 5.08 |
| 1823805 | 0.377 | 0.054 | 5.84 | 5.32 | 4.13 |
| 1920004 | 0.527 | 0.087 | 4.83 | 4.80 | 5.09 |
| 1921102 | 0.145 | 0.055 | 6.56 | 6.45 | 5.63 |
| 1922202 | 0.297 | 0.083 | 6.15 | 6.02 | 5.11 |
| 1922801 | 0.209 | 0.163 | 6.70 | 6.43 | 6.25 |
| 1926801 | 0.122 | 0.139 | 6.07 | 5.66 | 5.44 |

*Slope determined as $Abs_{365}(pH)/Abs_{365}(pH\ 1.5)$ vs. pH for each sample (see Fig. 3)






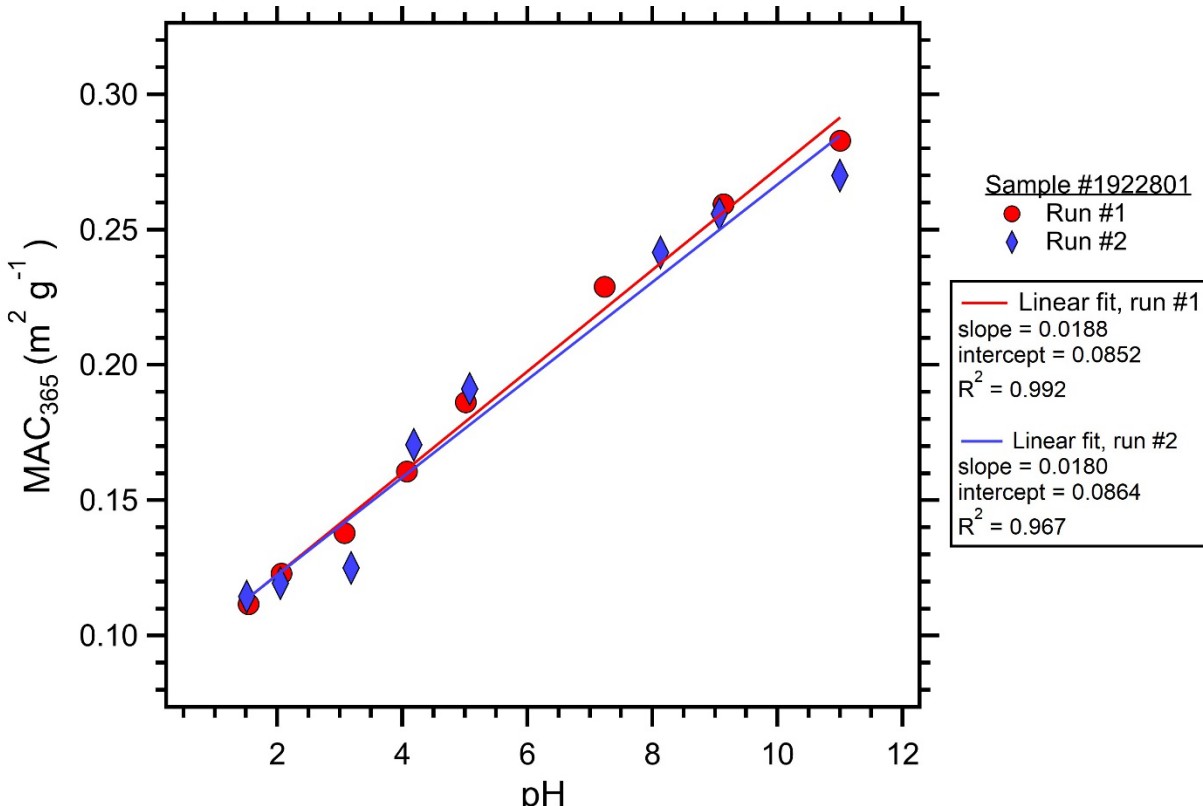

**Figure 1:** Duplicate measurements of MAC$_{365}$ as a function of pH for cloud water sample #1922801. The slope, intercept, and coefficient of variation (R$^2$) showed excellent repeatability for the analyses run for the same sample on different days. WSOC for this sample was 540 µM, close to the median value of 505 µM observed for this study.





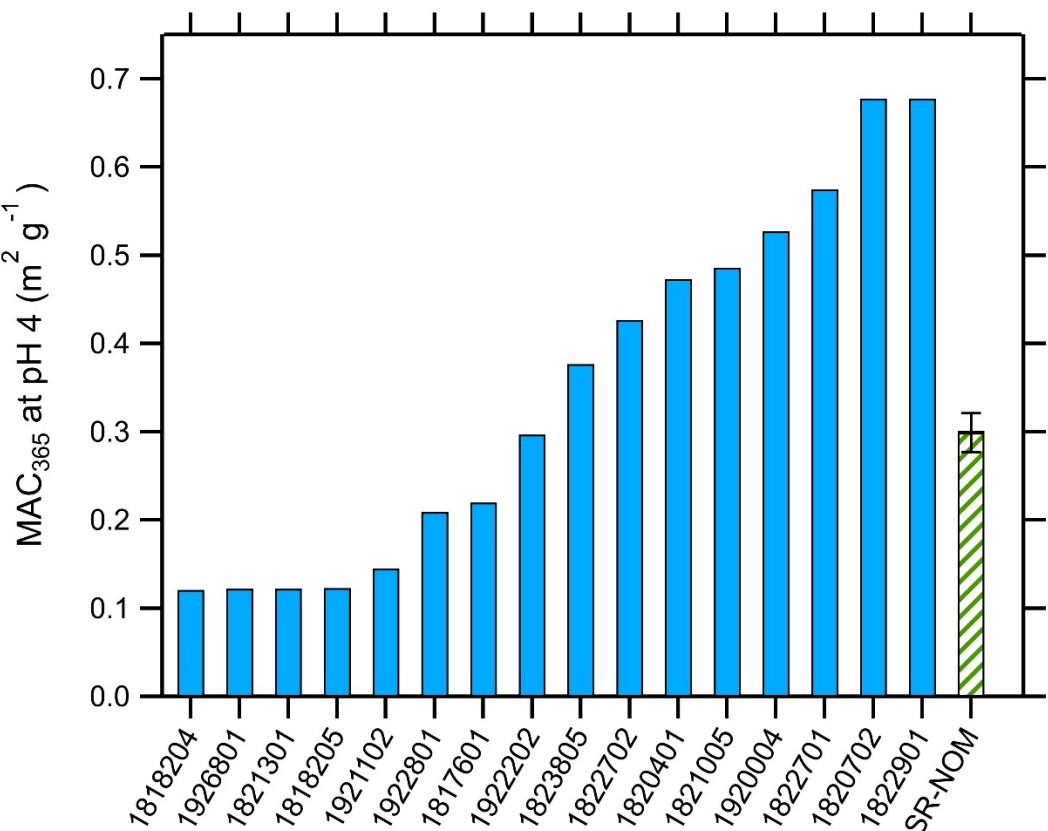

**Figure 2:** Average mass absorption coefficient measured at 365 nm ($MAC_{365}$) and pH 4 for cloud water samples (solid blue bars) and Suwannee River natural organic matter (hatched green bar).



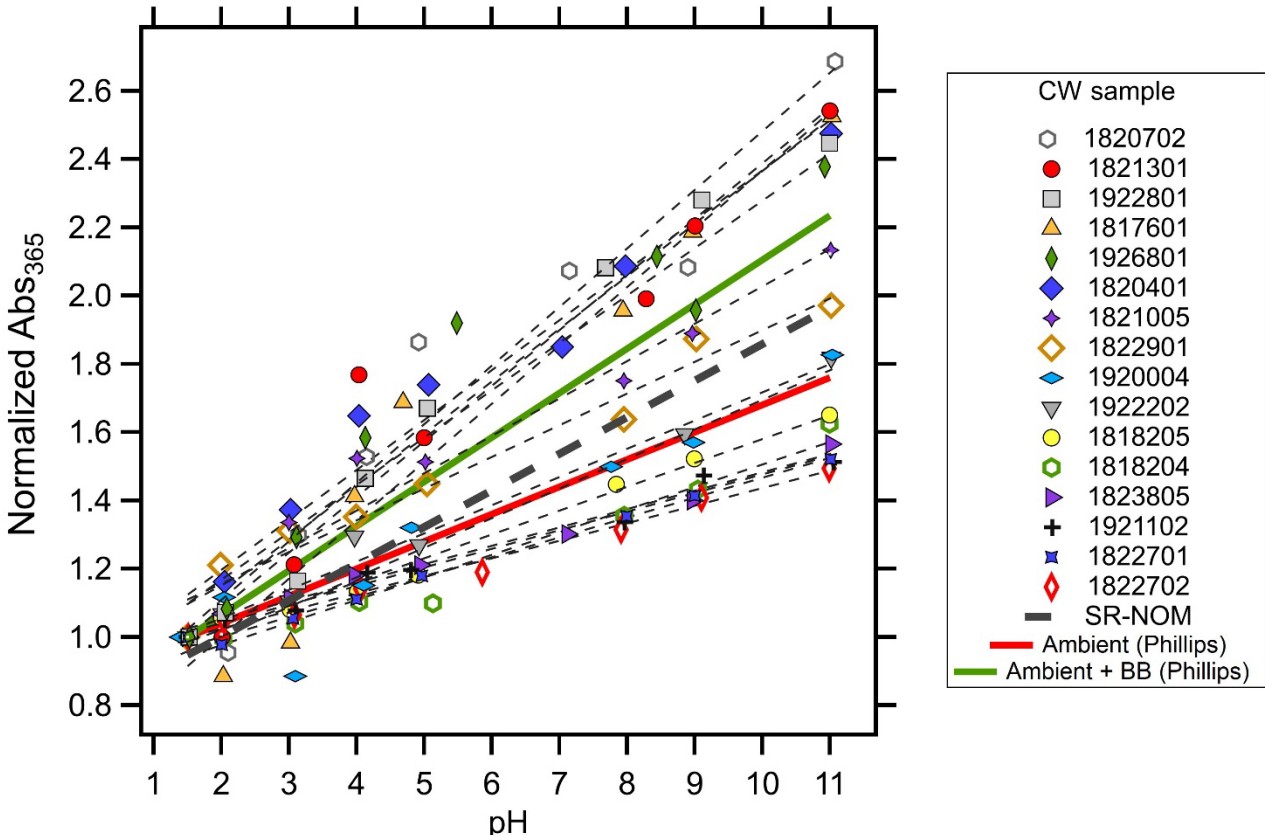

**Figure 3:** Normalized absorbance at 365 nm ($Abs_{365}(pH)/Abs_{365}(pH\ 1.5)$) versus pH for all cloud water samples analyzed in this study. Lines of best fit (least squares linear regression) for each cloud water sample are shown as the thin dotted lines. Comparison to BrC sampled in ambient aerosols both influenced and uninfluenced by biomass burning are shown with the solid green and red lines, respectively, both from Phillips et al. (2017). Mean behaviour of Suwannee River NOM shown with the thick dashed black line.





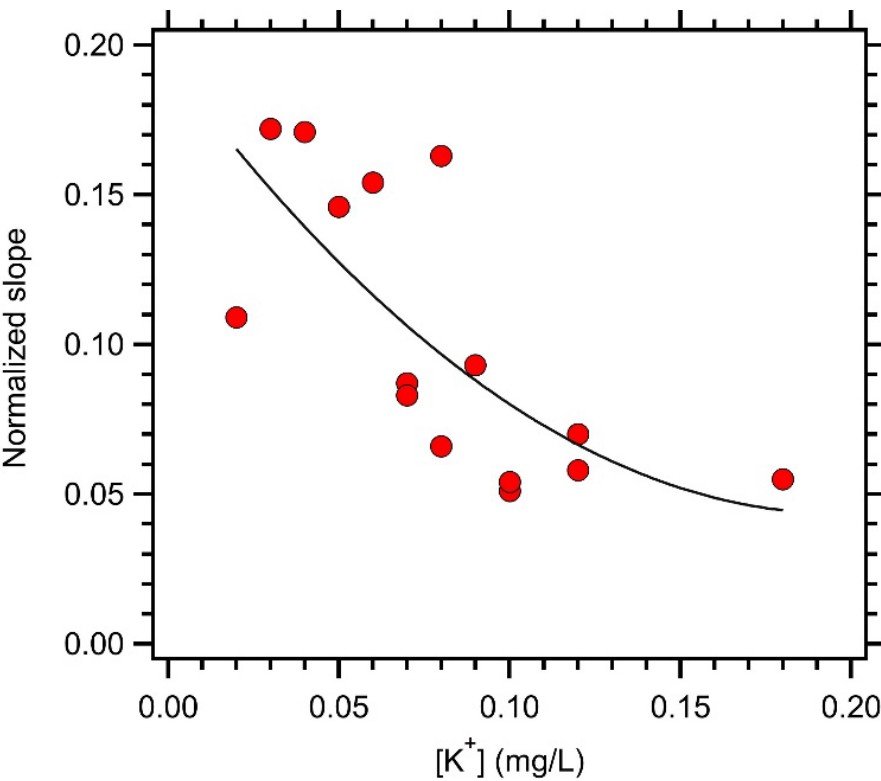

**Figure 4:** Relationship between the normalized slope of Abs$_{365}$ vs. pH (from Table 2) and the cloud water K$^+$ concentration.




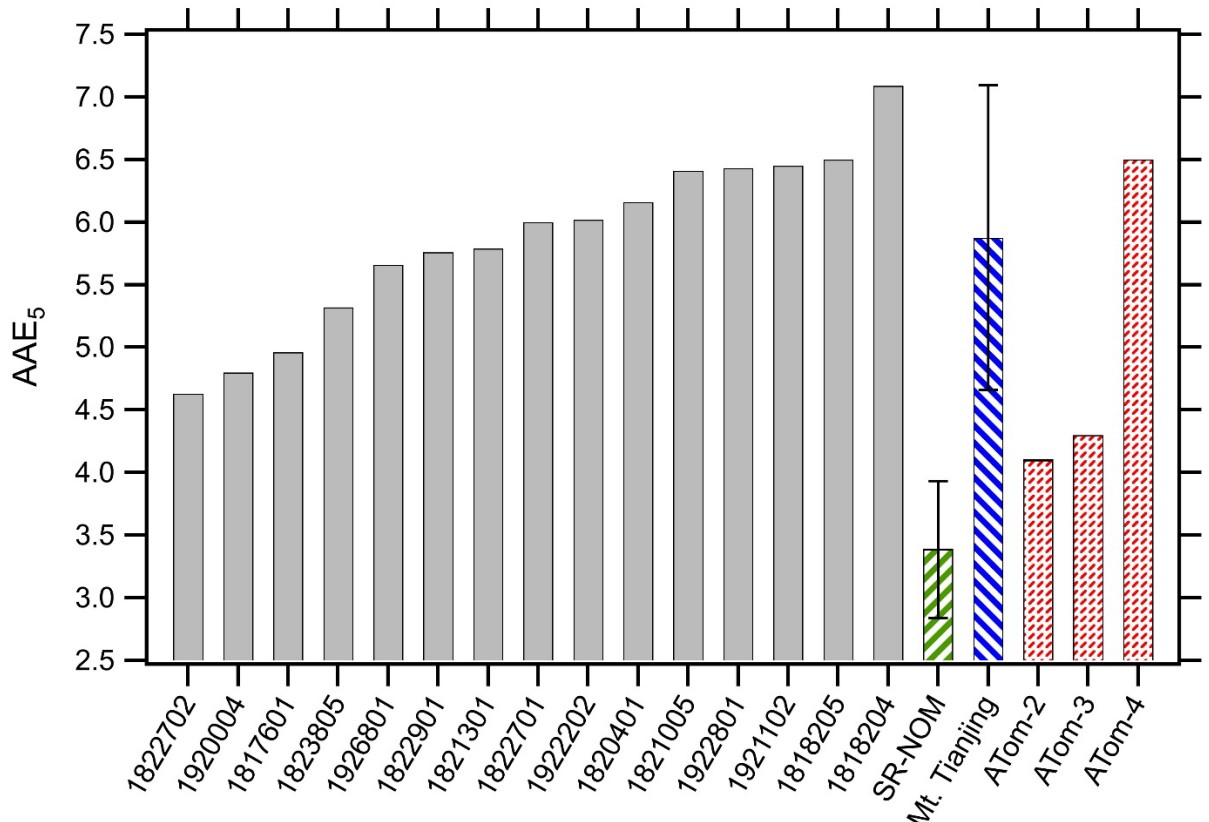

**Figure 5:** Distribution of absorption Ångström exponent values at pH 5 (AAE$_5$) for cloud water samples (solid bars), Suwannee River Natural Organic Matter (green hatched bar), cloud water samples from Mt. Tianjing, China (blue hatched bar, mean value ± 1σ reported from Guo et al., 2022 for WSOC), and mean values from the ATom aircraft missions (red patterned bars, from Zeng et al., GRL, 2020). Note, the pH corresponding to the AAE values from Mt. Tianjing and ATom missions were not specified.





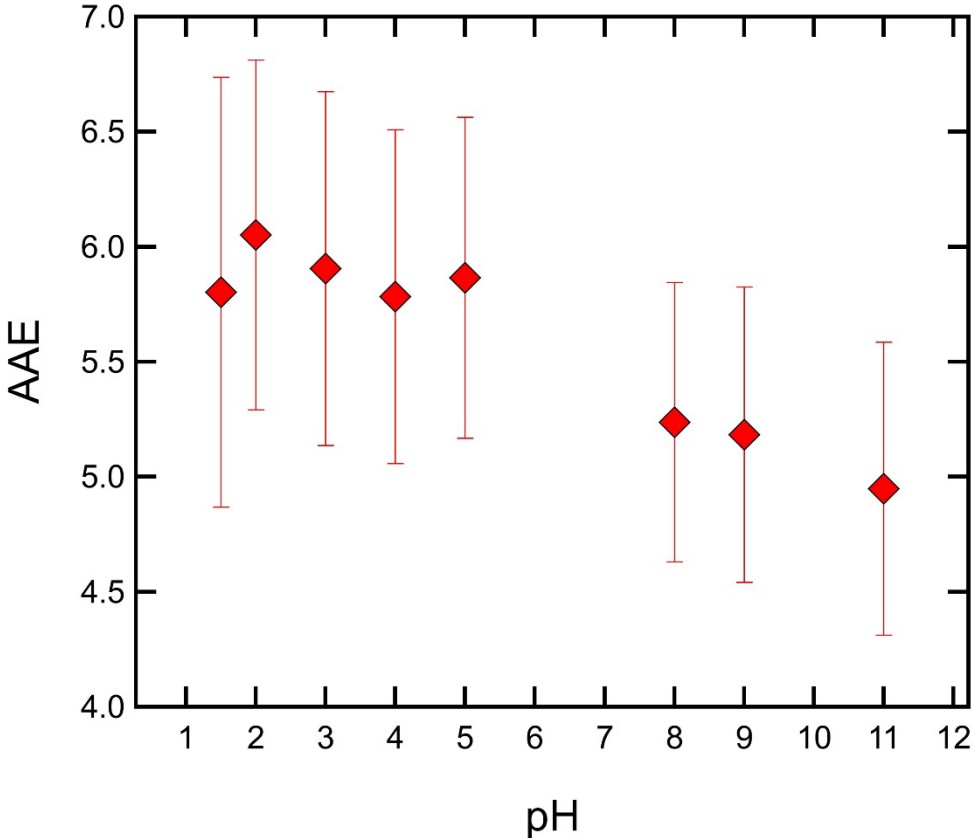

**Figure 6:** Mean absorption Ångström exponent value for all cloud water samples as a function of pH. Error bars represent ± 1σ.




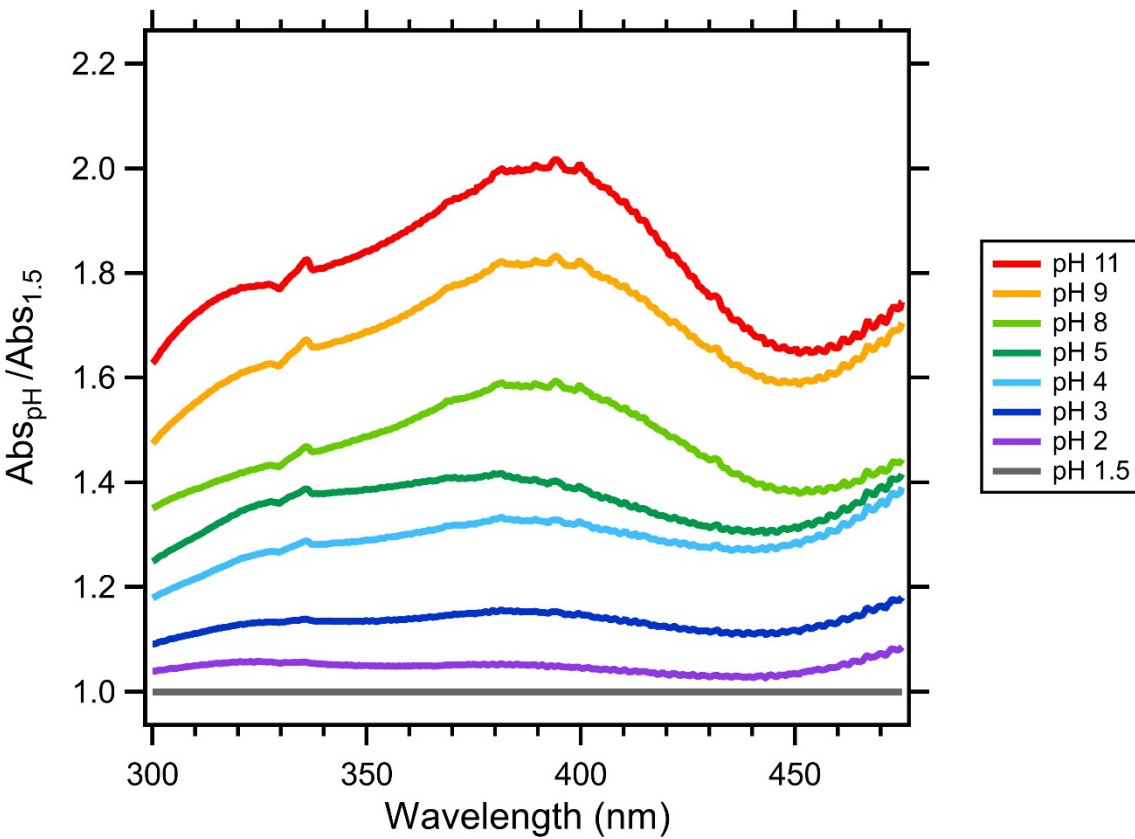

**Figure 7:** Mean absorption at a given pH relative to the absorption at pH 1.5 as a function of wavelength for all cloud water samples.


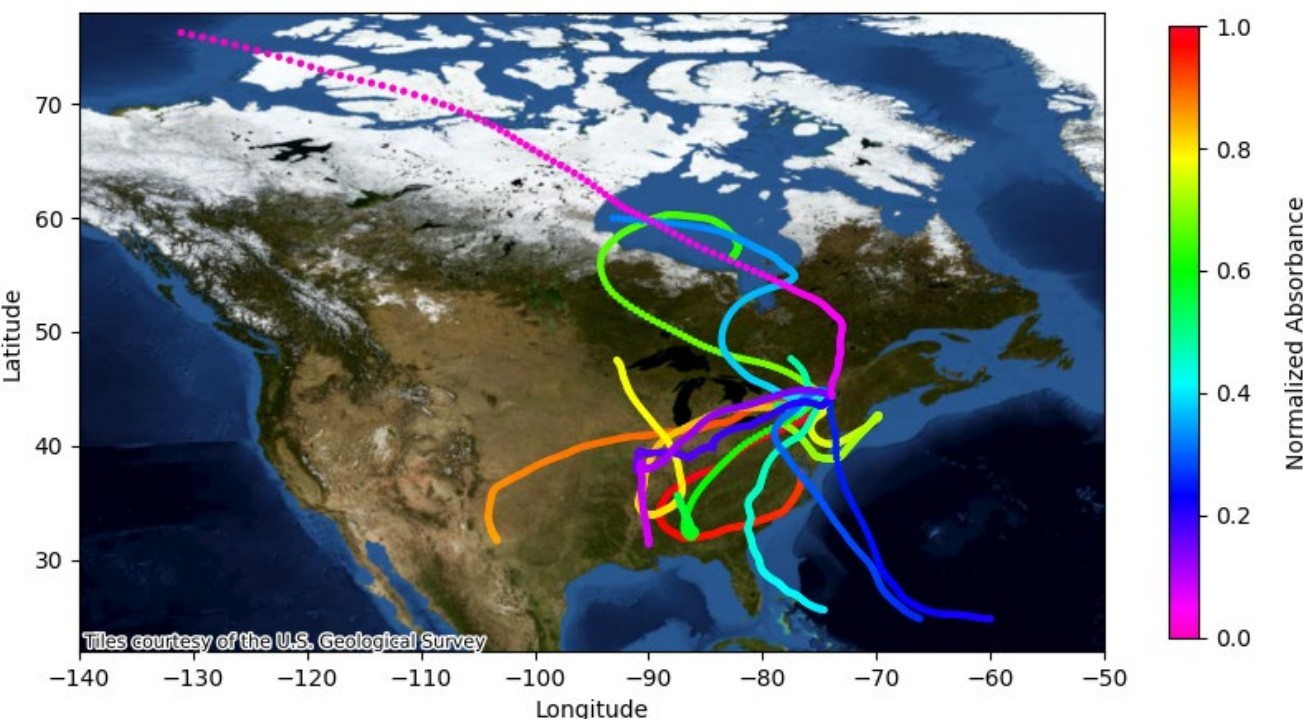

**Figure 8:** Map showing 6-day back trajectories for cloud water samples. Trajectories are coloured according to the relative BrC absorption at 365 nm (Abs$_{365}$) based upon the slopes (Table 2 and Fig. 3) and aerosol pH modelled along the back trajectory by ISORROPIA-II. The thermodynamic calculations assume constant aerosol composition along the back trajectory and only account for differences in pH based on T and RH.



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
