# Peer review of "pH-Dependence of Brown Carbon Optical Properties in Cloud Water"

_EGUsphere, 2023_

## Author Comment (AC1)

We thank the reviewers for their constructive comments. We have addressed each comment below, with the reviewer comment in black followed by our response in blue. We have also appended an updated manuscript after our responses with all changes highlighted.

**Referee #1**

This manuscript prepared by Hennigan et al. represents a novel study to provide insights into the impact of pH on the light absorptivity of atmospheric Brown Carbon (BrC). Through measurement of actual cloudwater samples, they have demonstrated that the light absorptivity of water-soluble BrC in cloudwater samples is highly pH dependent. While pH dependence of BrC absorptivity has been shown to some degree by previous studies, this work is the first to provide systematic insights into this phenomenon. BrC belongs to a class of short-lived climate forcers that has uncertain radiative effects and atmospheric lifetime. Meanwhile, a changing pH in cloud and fog water in North America and Europe has been reported. These facts make the current work highly relevant and important. The manuscript is very well written and should be considered for publication in ACP. I have the following comments and questions for the authors. I only have one major comment, with the rest considered minor or technical.

**Major comment**
- Generally, I think the manuscript can benefit from a little more discussion regarding potential mechanisms via which the observed pH dependence is attained. In Line 335, the authors mention hydrophobic organic acid. Do we expect this magnitude of changes in absorbance when the pH swing across their pKa value(s)? Have any previous studies on Suwanee River samples discussed how DOM exhibits pH dependence? I think such a mechanism is important in connecting a few key observations/conclusions in the manuscript: Ageing of BB aerosol, relative changes in absorption spectra, etc.

We agree with the reviewer's comment that more discussion of the underlying mechanisms is warranted. We have added the following text to the Discussion, (lines 294 – 304):

"The results in this study suggest that aromatic carboxylic acids and phenolic compounds, including nitrophenols, are primarily responsible for the observed pH-dependent optical properties. Phenols and aromatic carboxylic acids are major contributors to atmospheric BrC (Laskin et al., 2015). Prior studies have shown that aromatic carboxylic acids contribute most to the pH-dependence below pH 7 while phenols are most responsible above pH 7 (Schendorf et al., 2019; Qin et al., 2022). Our results also suggest that phenols are primarily responsible for the observed dependence of AAE on pH (**Figure 6**). AAE only varied with pH above pH 7, consistent with phenolic pKa values that are typically in the 7 – 10 range. The wavelength-dependent enhancements with increasing pH shown in **Figure 7** also point to the influence of phenols because the shape and magnitude of the enhancements become much more prominent above pH 7."

We also point the reviewer to several points in the manuscript where we discuss other studies that have observed a relationship between pH and BrC, including: lines 232 – 237 (updated manuscript line numbers), lines 273 – 276, and lines 356 – 367.

**Minor comments**
- As the authors pointed out themselves, aerosol liquid water represents a highly concentrated medium that is not considered ideal aqueous solution. Are acid-base equilibria, hence BrC absorption, differ in non-ideal solutions compared to ideal aqueous solutions? I do not know the answer. I am just asking.

This question has never been explored for atmospheric BrC; however, the effects of ionic strength on optical properties of BrC in aquatic and terrestrial environments (termed "chromophoric DOM") has been investigated (e.g., see Gao et al. (2015)). It is likely that ionic strength also affects BrC optical properties in aerosols and clouds, but it would be speculation at this point. This is a new topic of research that our group is pursuing.

- I wonder if the pH dependence observed in the current work is repeatable. In other words, if the authors would acidify the solution but then basify it again (or vice versa), do you expect the absorptivity to follow the same pH-dependence?

For a variety of different humic substances, including Suwannee River Humic Acid and Suwannee River Fulvic Acid, Schendorf et al. (2019) observed that the spectral changes with pH are completely reversible. This is consistent with aromatic carboxylic acids and phenols being the moieties primarily responsible for the pH-dependence we observe. We have added the following to the text (line 184 – 185) "Based on the results of Schendorf et al. (2019), we expect the observed pH-dependence to be reversible, although this was not verified experimentally."

- Related to my previous comment, certain aqueous-phase reactions (e.g., hydrolysis) and equilibria (imine formation) are acid and/or base-catalyzed. Is there any chance that acidifying or basifying the sample induces any irreversible artifact to the composition?

This is an excellent question. In theory – yes – pH changes could induce irreversible reactions of chromophoric WSOC in our cloud samples. However, we believe this is unlikely to occur in our samples because the LWCC measurements occur so soon after the pH adjustment (within a few minutes). Therefore, there is likely not enough time for any such reactions to proceed very far. To address this point, we have added the following text to line 129 – 130: "The sample was injected into the LWCC within minutes of the pH adjustment, minimizing the time for any acid-catalyzed reactions of BrC chromophores to occur."

- The authors mentioned that this study ignores water-insoluble BrC chromophores. Do the authors think water-insoluble chromophores also exhibit pH dependence? They do not dissolve or interact with water very much. I do not know the answer. I am just asking.

This is a very tricky question. The reason is that pH can be defined in non-aqueous media, such as organic solvents, but the pH scale between different solvents (including water) does not directly transfer (Himmel et al., 2010). To our knowledge, pH of the non-aqueous phase of

atmospheric organic aerosols has never been measured or estimated - given the complexity of OA in atmospheric particles, this would be quite daunting. Therefore, it is quite difficult to speculate about the potential pH behavior, given how little is known about the acidity of the non-aqueous phase of atmospheric OA.

- Page 5. It seems that the LWCC measurement was intentionally done using two channels, one to record absorbance and the other to track light source stability. How is this approach more advantageous compared to a single-channel measurement? I have seen previous studies using only one channel.

We implemented this measurement detail at the recommendation of Dr. Lelia Hawkins (personal communication). In our studies, our light sources have been extremely stable across many months of use. Eventually, the bulbs will have to be replaced so this measurement configuration will help us catch any changes early and without significant troubleshooting. However, we have not needed to apply any measurement corrections due to light source fluctuations to-date.

- Line 215. "The present results suggest that one such change not previously reported is that atmospheric ageing reduces the sensitivity of biomass burning BrC optical properties to pH ". I feel like this sentence is an overstatement and would ask the authors to consider relaxing the statement. I don't think the results really suggested it. It is the authors' speculation.

We agree with the reviewer's suggestion. We have edited the sentence so that it now reads: "The present results suggest that atmospheric ageing may reduce the sensitivity of biomass burning BrC optical properties to pH, though future studies are needed to substantiate this finding through investigations that provide for controlled ageing conditions."

- Line 254. "Therefore, measurements of BrC in aqueous environments need to include and report pH in order to facilitate interstudy comparisons and to assess the climate forcing effects of BrC." I agree with the authors and also believe that this is one of the most important implications they are making from this manuscript. Given that the authors demonstrated that the pH dependence also varies from sample to sample. Shouldn't we report absorbance at least two pH to constrain the slope? This may or may not always be feasible, but I wanted to hear the reviewer's ideas.

The reviewer is correct that reported absorbance values should include two pH levels, where possible. We have added the following to the text: "Further, optical properties of water-soluble BrC in aqueous environments should be measured at two pH levels, when feasible, to enable translation to other conditions in the atmosphere."

- Discussion related to Figure 7. I think it would be beneficial if the authors could include a little discussion on what functional groups are likely (or unlikely) contributing to the observed relative spectral change. E.g., carbonyl is not likely contributing due to minimal changes at around 300 nm.

See our reply to this reviewer's major comment above.

**Technical Comment**

- I feel like the use of Figure X and Fig. X is inconsistent throughout the manuscript.

To be consistent we have now used 'Figure X' throughout the manuscript.

**Referee #3**

This paper describes observations of the absorption characteristics of brown carbon in cloud water samples obtained from orographic clouds at Whiteface Mountain. Recent work has demonstrated that brown carbon absorption is highly pH dependent, however, the number of studies that report optical properties in aqueous samples as a function of pH is limited. Furthermore, the majority of these studies have analysed aerosol samples in aqueous solution but cloudwater is more dilute, typically has a higher pH and therefore demonstrates stronger absorption. However, to date few studies exist. This work presents such a set of observations. The work is well described and the methods carefully detailed in the paper. The results are carefully described and logically presented and the authors have provided a well considered and detailed discussion, comparing their results with those of other studies and discussing the ramifications of their work. Overall, this is a well presented paper that is of merit scientifically and offers some new results and important insight. I have some small suggestions that the authors should address, but these are minor.

Line 46-49: "Unlike BC and dust, which are removed from the atmosphere only through wet and dry deposition, it also undergoes chemical losses initiated by oxidants and direct photolysis (collectively termed bleaching), that can rapidly diminish its light absorbing properties (Hems and Abbatt, 2018)". This sentence needs a re-word. The processes discussed for BrC are in addition to the physical processes controlling BC and dust.

The sentence has been reworded so that it now reads: "Like BC and dust, BrC is removed from the atmosphere through wet and dry deposition; however, BrC also undergoes chemical losses initiated by oxidants and direct photolysis (collectively termed bleaching) that can rapidly diminish its light absorbing properties (Hems and Abbatt, 2018)."

It is worth emphasising in the introduction as well as in the conclusions that many studies only focus on the optical properties of brown carbon under dry conditions and also that multiple studies considered absorption at ambient humidity have not reported the aerosol pH at which the determinations have been made.

This is an excellent point. The effects of drying on the optical properties of BrC are unknown; however, the transition of pH as ambient particles are dried, as occurs with many widely used aerosol measurement systems, will change the water-soluble BrC absorption. We have added the following text to the Discussion: "These results also inform measurements of BrC that are not conducted in aqueous matrices. For example, experimental approaches such as cavity ringdown spectroscopy (CRDS), photoacoustic spectrometry (PAS), and aethalometry are frequently used to measure total BrC, not just the water-soluble fraction (Liu et al., 2015). Non-filter based approaches, including CRDS and PAS, typically dry the air sample before measurement

(Washenfelder et al., 2013; Lack et al., 2012).  It is unclear how the optical properties of chromophoric WSOC change as the particles transition from ambient conditions, where they often contain liquid water, to the dry environment within the instrument.  Our results suggest that the water-soluble BrC compounds that exhibit a pH dependence will also exhibit different absorbance behaviors transitioning from an aqueous to non-aqueous phase state, though this topic should be explored in detail in the future."

Line 95-97: How were non-precipitating clouds selected?

Schwab et al. (2016) indicates that cloud water collection occurs when: "the heated grid rain sensor must indicate that no rain is present, to assure that samples are from non-precipitating clouds."

Line 147-149 and table 1: "with the exception that $Ca^{2+}$ and $Mg^{2+}$ concentrations were excluded because a decadal analysis of WFM cloud composition revealed that these species likely derive predominantly from coarse particles" If these cations are predominately in the coarse mode, which anions correspond to the coarse mode?  How can it be discounted that the $Na^+$, $Cl^-$ and $NO_3^-$ do not have a significant coarse mode contribution? What role would this play on the aerosol pH calculation?

Lawrence et al. (2023) have a very detailed discussion and analysis of this point – see especially their Section 6.3.  While the reviewer correctly points out that other species are present in the course mode, as well, their imbalance between number and mass fractions is not nearly as disparate as it is for $Ca^{2+}$ and $Mg^{2+}$.  Lawrence et al. show convincingly that $Ca^{2+}$ and $Mg^{2+}$, likely balanced with unmeasured species like carbonate, are overrepresented in the cloud water compared to the aerosol.

Line 155-159: To what extent does the Mountain affect airflow and therefore the ability of HYSPLIT capture the airmass history accurately?

We agree with the reviewer that this could be a concern for the 1-2 hours immediately before sampling.  However, because our trajectory analysis extended backwards for 144 hours, the uncertainty imposed by the orographic effects should be relatively minor overall.

Lines 195-196: The authors comment on the greater variability in their results compared to the aerosol measurements of Phillips et al and suggest this may be due to ageing of air masses at the sample site.  Another plausible explanation is that the activation characteristics of the aerosol are a source of variability.  Unlike Phillips et al, who studied aerosol, the cloud water samples only observe activated aerosols.  Since this is a strong function of both the aerosol size distribution and the updraft velocity and BrC is likely to be prevalent in the unactivated aerosol, greater variability may be induced in the observations.  This is worth commentary.

We agree with the reviewer that the activated fraction of BrC-containing particles has importance for the pH of the aqueous environment and optical properties of BrC - see our reply to the next comment and our addition to the text.  However, in this case, Phillips et al. (2017) have collected ambient aerosols onto filters and extracted the BrC in water, and filtered the

extracts.  Therefore, their study also only considered the water-soluble fraction of BrC, similar to ours.

Lines 280-284: This discussion also implies that it is also important to quantify the available activated fraction of BrC from a range of important sources as a function of age.

The reviewer brings up an excellent point.  To our knowledge, no studies have examined the activated fraction of BrC, but this is certainly worth exploring in light of our present results.  We have added the following text to this discussion:
"The activated fraction of BrC has, to our knowledge, never before been explored but also affects the radiative forcing of BrC in aerosols and clouds."

[revised manuscript text omitted]